# Prevalence and factors associated with physical function limitation in older West African people living with HIV

Charlotte Bernard[1,2]*, Hélène Font[1,2], Zélica Diallo[3], Richard Ahonon[4], Judicaël Malick Tine[5], Franklin Abouo[3], Aristophane Tanon[3], Eugène Messou[4], Moussa Seydi[5], François Dabis[1,2], Nathalie de Rekeneire[1,2], The IeDEA West Africa Cohort Collaboration[¶]

1 Univ. Bordeaux, INSERM, Bordeaux Population Health Research Center, UMR 1219, Bordeaux, France, 2 Univ. Bordeaux, ISPED, Bordeaux Population Health Research Center, UMR 1219, Bordeaux, France, 3 Service de maladies infectieuses et tropicales, CHU Treichville, Abidjan, Côte d'Ivoire, 4 Centre de prise en charge de recherche et de formation (CePReF), Yopougon Attié Hospital, Abidjan, Côte d'Ivoire, 5 Service de maladies infectieuses et tropicales, CRCF, CHNU de Fann, Dakar, Senegal

¶ Complete membership of the author group can be found in the Acknowledgments
* charlotte.bernard@u-bordeaux.fr

**Data Availability Statement:** All relevant data are uploaded to Figshare and publicly accessible via the following DOI: 10.6084/m9.figshare.12431078.

## Abstract

Although physical function decline is common with aging, the burden of this impairment remains underestimated in patients living with HIV (PLHIV), particularly in the older people receiving antiretroviral treatment (ART) and living in sub-Saharan Africa (SSA). PLHIV aged ≥50 years old and on ART since ≥6 months were included (N = 333) from three clinics (two in Côte d'Ivoire, one in Senegal) participating in the International epidemiological Databases to Evaluate AIDS (IeDEA) West Africa collaboration. Physical function was measured using the Short Physical Performance Battery (SPPB), the unipodal balance test and self-reported questionnaires. Grip strength was also assessed. Logistic regression was used to identify the factors associated with SPPB performance specifically. Median age was 57 (54–61) years, 57.7% were female and 82.7% had an undetectable viral load. The mean SPPB score was 10.2 ±1.8. Almost 30% had low SPPB performance with the 5-sit-to-stand test being the most altered subtest (64%). PLHIV with low SPPB performance also had significantly low performance on the unipodal balance test (54.2%, p = 0.001) and low mean grip strength (but only in men (p = 0.005)). They also showed some difficulties in daily life activities (climbing stairs, walking one block, both p<0.0001). Age ≥60 years (adjusted OR (aOR) = 3.4; CI95% = 1.9–5.9,), being a female (aOR = 2.1; CI95% = 1.1–4.1), having an abdominal obesity (aOR = 2.1; CI95% = 1.2–4.0), a longer duration of HIV infection (aOR = 2.9; CI95% = 1.5–5.7), old Nucleoside reverse transcriptase inhibitors (NRTIs) (i.e., AZT: zidovudine, ddI: didanosine, DDC: zalcitabine, D4T: stavudine) in current ART (aOR = 2.0 CI95% = 1.1–3.7) were associated with low SPPB performance. As in western countries, physical function limitation is now part of the burden of HIV disease complications of older PLHIV living in West Africa, putting this population at risk for disability. How to screen those impairments and integrate their management in the standards of care should be investigated, and specific research on developing adapted daily physical activity program might be conducted.

Additional inquiries may be sent to the authors at charlotte.bernard@u-bordeaux.fr.

**Funding:** Supported by the National Institute of Mental Health (NIMH), National Cancer Institute (NCI), the Eunice Kennedy Shriver National Institute of Child Health & Human Development (NICHD) and the National Institute of Allergy and Infectious Diseases (NIAID) of the U.S. National Institutes of Health (NIH), as part of the International Epidemiologic Databases to Evaluate AIDS (IeDEA) under Award Number U01AI069919. The content is solely the responsibility of the authors and does not necessarily represent the official views of the National Institutes of Health.

**Competing interests:** NO authors have competing interests.

## Introduction

According to UNAIDS, 6.7 million people aged 50 and older were living with HIV worldwide in 2017, a phenomenon that increased steadily since 1995 [1]. Sub-Saharan Africa (SSA) represents the region with the largest number of older people living with HIV (PLHIV) (4.1 million) [1]. With massive and rapid access to antiretroviral therapy (ART), the overall improvement in HIV medical care services, and the dramatic change in the demographics of the HIV-infected population, HIV infection can now be considered as a chronic disease worldwide. Older patients receiving ART are thus an increasing population group and have an increased risk of developing age-associated non-communicable comorbidities [2,3].

With aging, alteration of physical function (i.e. functional limitation and disability) are more common. Independence and social life are clearly related to physical function in older adults [4]. A functional limitation is a strong predictor of disability, hospitalization, nursing home admission, and mortality in aging [5,6]. In the context of HIV infection, this problem has long been neglected although up to 30% of PLHIV in Canada, for instance, have complaints about their physical function [7]. Recent US data showed that PLHIV are affected by those impairments even at middle age [8]. In France, it has been reported that one of two middle-aged PLHIV had poor limb muscle performance [9], that the performance deteriorated over time and that was associated with subsequent falls [10]. In Cote d'Ivoire, one study reported a higher prevalence of physical function limitation (34%) in middle-aged PLHIV, with older age and higher BMI being associated with low performance [11].

As those impairments emerge as an important public health issue, there is a risk of a significant burden on the healthcare systems and human resources, especially in SSA. Few data are available in SSA, particularly in PLHIV aged 50 years and above. The aim of our study was to describe the prevalence and identify factors associated with physical function impairment in PLHIV aged 50 years and above living in West Africa.

## Methods

### Study population

This study is a part of an ancillary project within the West Africa network of the International epidemiological Databases to Evaluate AIDS (IeDEA) of the US National Institutes of Health (https://www.iedea.org/regions/west-africa/) [12]. This cross-sectional analysis is part of a 2-year longitudinal study evaluating different aspects of aging with HIV (cognition, physical function, depression and frailty) with a follow-up still ongoing. By convenience, the study was conducted in three urban clinics with a large case load of PLHIV in two different countries: the infectious and tropical disease department of the Treichville University Hospital, the referral public clinic (CePReF) in Yopougon Attié Hospital (Abidjan, Côte d'Ivoire), and the infectious and tropical disease department of the Fann University Hospital (Dakar, Senegal). The inclusion period of the study was between February 2016 to November 2017.

Patients were eligible if they were adults living with HIV-1, 50 years old or older, and on ART for at least 6 months. We excluded patients having history of cerebral opportunistic infection, neurological pathology (history of stroke, Parkinson disease), current disabling opportunistic infection, meningitis, sensitivo-motor paralysis, psychiatric illness (including psychotropic treatment), cancer under treatment or respiratory or cardiac insufficiency.

The national ethics committee of each participating country approved the protocol (Senegal: Conseil National d'Ethique de la Recherche en Santé (CNERS), protocol SEN15/74; Côte d'Ivoire: Comité National de l'Ethique et de la Recherche, protocol n°009/IMSLS/CNER-k). All the patients gave their written consent before being included in the study.

## Socio-demographic and medical data

Patients' characteristics including information on socio-demographics, anthropometry, medical history, HIV clinical data, and also substance use (tobacco, drugs, and alcohol with the AUDIT-C (a score ≥3 for women or ≥4 for men was considered as hazardous drinking) have been collected through basic questions and medical examination.

Professional activity was defined in two categories: employed and unemployed. The Body Mass Index (BMI) was calculated by dividing the weight (in kilograms) by the height (in meters squared) and divided into three categories: overweight/obesity when BMI ≥25 kg/m$^2$, normal when $18 \leq$ BMI $<25$ and low when $<18$ kg/m$^2$. Abdominal obesity was defined by an abdominal circumference ≥80 cm in women and ≥94 in men [13].

Concerning medical data, patients were asked if they have ever been diagnosed with any chronic diseases (hypertension, diabetes, hyperlipidemia, migraine, arthrosis, tuberculosis, etc.). The initial clinical stage using Centers for Disease Control and Prevention (CDC) definition was used (A, B, or C) [14]. Exposure to zidovudine (AZT), didanosine (ddI), stavudine (D4T), zalcitabine (DDC) in the initial and the current ART treatment was studied through a categorical variable (yes/no). Adherence to ART was defined as the percentage of tablets the patient declared to have taken over seven days (in comparison to the prescribed total number of tablets over this period).

## Physical function limitation

Physical function limitation was evaluated with different tests of physical performance, and by self-reported measures. These tests were conducted by trained staff following standard procedures. Each time was recorded with a digital stopwatch.

## Physical performance

■ The Short Physical Performance Battery (SPPB) [15] assesses low extremity function using three timed components in the following order: balance, gait speed, and low limb strength (5 times sit-to-stand test - 5STS). A score between 0 to 4 was attributed to each subtest, depending on the performance as defined in the Guralnik et al. standard procedures [15]. The sum of the three components comprised the final SPPB score with a possible range from 0 to 12 (12 indicating the highest degree of low extremity functioning). For physical limitation, as defined by Guralnik et al [15], poor physical performance was defined as a total SPPB score ≤9. To describe in detail which SPPB subtest was the most altered, we used these following cutoffs: a score <4, except for walking speed (the cutoff < = 0.8m/s was used [16]).

■ The unipodal balance test [17] consists of standing on one leg as long as possible (max 30 seconds), with open eyes. The test was repeated two times. The best performance from the 2 tests was used for the analyses. The unipodal balance test was considered as failed when the patient could not stand this position for 30 seconds.

■ A "physical function impairment" variable was also described. A patient was considered as having physical function impairment if at least more than one test (i.e. balance, gait speed, 5-STS or unipodal balance test) was altered, as defined above in the text for each test.

**Self-reported measures.** Additionally, daily physical function tasks were evaluated based on self-reported difficulties in daily physical function such as climbing stairs, raising arm, carrying a 5kg grocery bag, walking a long distance, participating in community activities, and visiting family. Each difficulty was evaluated with a Likert scale (no difficulty / mild / moderate / severe / extreme difficulties or cannot do).

## Grip strength

As SPPB assesses low extremity function and to have a complete description of muscular function in our PLHIV, the handgrip strength (kg) was measured to evaluate upper extremity maximal isometric muscle strength. The measure was realized in a standard way, twice consecutively in the dominant hand using a calibrated Jamar hydraulic hand dynamometer. The maximum value of the 2 trials was used for the analysis. For grip strength, as no normative data obtained from a population with similar characteristics to ours were available, low grip strength was defined as a performance less to the median obtained in each gender group (male/female).

## Measures of functional status

Activities of the daily living (ADL) [18] and Instrumental activities with daily living (IADL) [19] scales were used to evaluate the autonomy of the patients. The ADL and IADL scores range from 0 to 6 and 0 to 4, respectively (0 indicating the lowest degree of autonomy).

## Other covariates

The practice of physical activity was evaluated with the WHO Global Physical Activity Questionnaire (GPAQ) [20], collecting information on physical activity in three situations: at work, during moving from one place to another, and during hobbies. The last question evaluates physical inactivity (how many hours do you spend sitting or lying down during a day (without sleeping). For the present study, we focused on global scores: physical inactivity (the patient did not do physical activities at work or in leisure: yes/no) and intensity of physical activity (limited to moderate vs high) which were calculated following the GPAQ analysis guide instructions.

## Statistical analysis

The characteristics of the study sample were described with numbers and proportions as variables were presented in categorical variables. The median and interquartile range (IQR) for describing the age and duration of the disease variables were also provided. For each physical function tests, mean scores and standard deviation were provided. Grip strength was only described according to gender.

For each test and the "physical function impairment" variable, the prevalence of low performance was reported. As SPPB has been used for >20 years to measure physical performance in older adults [15] and as it is a complete brief battery feasible to implement in HIV clinical care [21], the further analyses focused on this test. Associations between low SPPB performance and both other tests performance and daily physical activities were evaluated with Chi-2 tests, except for grip strength for which the Student T-test was performed in each gender group separately.

Factors associated with low SPPB performance were evaluated with univariable and multivariable logistic regression analyses. The multivariable logistic regression models included all variables associated with the dependent variable with a p-value ≤0.2 in univariable analyses. Unbalanced variables (85%/15%) were excluded from the analyses. The final model was obtained with a backward selection, and we considered significant associations at $p < 0.05$. "Inclusion centers" variable was included as a cofounder in each model. The goodness of fit of the final model was evaluated with the Hosmer-Lemeshow test ($p>0.05$).

A multivariable imputation of missing data was performed with a Random Forest procedure. No significant difference was observed between the two databases (with and without missing data). The database with imputed missing data was used for the logistic regression analyses.

Statistical analyses were computed using R software.

## Results

### Characteristics of the sample

A total of 333 patients were included in our study. The median (IQR) age was 57 (54–61) years old. Among them, 35.1% were aged 60 years and older, 57.7% were female, and 50.7% had a primary school or a lower level of education. Half of them lived alone (53.7%) and almost half of them were unemployed (46.8%) (Table 1).

Among these patients, 53.1% had abdominal obesity and 37.8% were overweight or obese. Less than a quarter ever had hypertension (21.9%) and/or tuberculosis (21.6%). Other comorbidities (diabetes, hyperlipidemia, migraine, and arthrosis) were less prevalent (6%, 3.6%, 10.2% and, 15.6%, respectively).

The majority of patients had an undetectable viral load (68.8%), half of them had CD4≥500 (50.4%) and 60.9% had a Nadir CD4 <200. The median (IQR) duration of HIV infection was 108 months (68.9–141.4). Fourteen percent (14.4%) were on stage C at ART initiation. Concerning ART, 68.2% had AZT, ddI, D4T, DDC included in their initial treatment whereas only 26.7% had those molecules in their current treatment.

Few patients reported substance use (hazardous drinkers or drug users <8%, except 17.7% for tobacco users (current/previous)).

In terms of the ADL and IADL instruments, 96.4% and 99.1% of the patients obtained the maximum score (6 or 4, respectively). Overall, 44.4% reported being physically inactive and, among those reporting being active, 53.4% reported limited to moderate intensity in their physical activity (data not shown).

### Description of physical function, self-reported measures, and grip strength

The SPPB mean score was 10.2±1.8. Details in SPPB total and subscores are presented in supplementary data (S1 and S2 Figs). The mean time to perform the 5STS was 12.5 ±3.5 seconds. The mean walking speed was 1.3 ±0.3 meters per second.

Few patients reported moderate to extreme difficulties in daily physical function (<4%), except for climbing stairs (14.1%) and carrying a grocery bag (8.4%) (S1 Table). Among patients reporting those difficulties, few of them reported severe or extreme difficulties (<2.5%). Only 5.4% of the patients reported at least one fall in the past year.

For women, the mean grip strength was 24.7 ±7.7 kg with 43.2% showing low grip strength whereas, for men, the mean grip strength was 40.8 ±12.3 kg with 42.5% having low grip strength.

### Prevalence of physical function impairment

Almost one-third of the patients had low SPPB performance (28.8%; 95% Confident Interval (CI): 24.0–33.6) (Table 2). Among the components of the SPPB, the 5STS was the most altered test (64.2% [59.0–69.3] had a score of less than 4) whereas less than 27% had low performance for balance or walking speed. Concerning unipodal balance, 38.6% had low performance. Overall, 45.6% [40.2–50.9] of the patients had physical function impairment.

### Associations between SPPB performance and other tests

A significant association between SPPB score and unipodal test performance was observed ($\chi 2 = 10.7$, p = 0.001). Of 96 PLHIV with low SPPB performance, 54.2% had low performance on unipodal test. Concerning grip strength, men with low SPPB performance had significantly

**Table 1. Characteristics of the study sample (N = 333).**

| Characteristics | Number | Percentages (%) |
|---|---|---|
| **Socio-demographic data** | | |
| Age (years) | | |
| 50–59 | 216 | 64.9 |
| 60 et + | 117 | 35.1 |
| Gender | | |
| Male | 141 | 42.3 |
| Female | 192 | 57.7 |
| Marital status | | |
| In couple | 154 | 46.2 |
| Alone | 179 | 53.7 |
| Level of education | | |
| Primary or less | 169 | 50.7 |
| Secondary or more | 164 | 49.2 |
| Professional activity | | |
| Employed | 177 | 53.1 |
| Not employed | 156 | 46.8 |
| **Anthropometric and medical data** | | |
| Abdominal obesity (mis. 1) | 177 | 53.1 |
| Overweight/obesity (mis. 1) | 126 | 37.8 |
| Hypertension (mis. 1) | 73 | 21.9 |
| History of tuberculosis | 72 | 21.6 |
| Arthrosis (mis. 1) | 52 | 15.6 |
| History of neurological disease | 47 | 14.1 |
| Migraine | 34 | 10.2 |
| History of trauma (mis. 2) | 22 | 6.6 |
| Diabetes | 20 | 6.0 |
| Other medical problem (mis. 3) | 14 | 4.2 |
| Hyperlipidemia (mis. 2) | 12 | 3.6 |
| B or C hepatitis (mis. 1) | 9 | 2.7 |
| **HIV Clinical data** | | |
| Duration of infection (months) | | |
| [7.29,88.4] | 111 | 33.3 |
| (88.4,131] | 112 | 33.6 |
| (131,317] | 110 | 33.1 |
| Clinical disease stages at ART initiation | | |
| A | 100 | 30.1 |
| B | 181 | 54.3 |
| C | 48 | 14.4 |
| Missing | 4 | 1.2 |
| Nadir CD4 (cells/μl) | | |
| <200 | 203 | 60.9 |
| ≥200 | 120 | 36.1 |
| Missing | 10 | 3.0 |
| More recent CD4 (cells/μl) | | |
| <500 | 162 | 48.6 |
| ≥500 | 168 | 50.4 |
| Missing | 3 | 1.0 |

(*Continued*)

**Table 1.** (Continued)

| Characteristics | Number | Percentages (%) |
|---|---|---|
| Detectable Viral load | 48 | 14.4 |
| Missing | 56 | 16.8 |
| Initial treatment including AZT, ddI, D4T, DDC† | 227 | 68.2 |
| Current treatment including AZT, ddI, D4T, DDC | 85 | 25.5 |
| Poor Adherence | 19 | 5.7 |
| **Substance use** | | |
| Hazardous drinkers | 24 | 7.2 |
| Tobacco use (current/former) | 59 | 17.7 |
| Drug consumption (mis. 1) | 6 | 1.8 |
| **Physical function** Mean performance ± SD * | | |
| SPPB (score) | 10.2 ±1.8 | |
| 5STS (time–seconds) | 12.5 ±3.5 | |
| Walking speed (m/s) | 1.3 ±0.3 | |
| **Activities of daily living (score = 6)** | 321 | 96.4 |
| **Instrumental Activities of daily living (score = 4)** | 330 | 99.1 |
| **Physical inactivity** | 148 | 44.4 |

**Abbreviations:** ART: Antiretroviral, AZT: Zidovudine, ddI: Didanosine, DDC: Zalcitabine, D4T: Stavudine, mis.: Missing, m/s: Meter per second, SPPB: Short Physical Performance Battery, 5STS: Five Sit-To-Stand, SD: Standard deviation.

*Unipodal test results are presented in Table 2 only (categorical variable).

† Others drugs included in ART could be: 3TC: Lamivudine, ABC: Abacavir, ATV: Atazanavir, DRV: Darunavir, EFV: Efavirenz, FTC: Emtricitabine, LPV: Lopinavir, NVP: Névirapine, RTV: Ritonavir or TDF: Ténofovir.

lower mean grip strength than men with high SPPB performance (35.7±9.1 kg vs 41.9±12.7, T = 2.94, p = 0.005). No significant difference was observed between both women's groups (T = 0.94, p = 0.93).

## SPPB performance and daily physical activities

A significant association was observed between low SPPB performance and physical inactivity (57.3%, χ2 = 8.3, p = 0.004). Low SPPB performance was also associated with limited to moderate intense activities (61.5%, χ2 = 11.2, p = 0.001). Concerning daily physical activities, low

**Table 2.** Prevalence of physical function impairment for each test.

| Tests | Prevalence (%) [95%CI]** |
|---|---|
| SPPB (score ≤9) | 28.8 [24.0–33.7] |
| SPPB subtests | |
| 5STS (score<4) | 64.2 [59.0–69.3] |
| Walking speed (speed < = 0.8m/s) | 27.0 [22.2–31.8] |
| Balance (score <4) | 16.2 [12.2–20.2] |
| Unipodal balance (Vereeck et al, 2008) | 38.6 [33.4–43.8] |
| Physical function impairment | 45.6 [40.2–50.9] |

**Abbreviations:** CI: Confident Interval, SPPB: Short Physical Performance Battery, 5STS: Five Sit-to-Stand, m/s: meter per second.

** The prevalence data is for the patients who scored below the cut off values.

SPPB performance was also associated with difficulties in climbing stairs (45.8%, χ2 = 13.7) and walking one block (22.1%, χ2 = 14.6) (p<0.0001 for both).

## Factors associated with low SPPB performance

In univariable models (Table 3), age ≥60 years (OR = 2.7; CI95% = 1.6–4.5), being a female (OR = 2.2; CI95% = 1.3–3.7), single (OR = 1.8; CI95% = 1.1–3.0) and unemployed (OR = 2.4; CI95% = 1.4–3.9) were associated with low SPPB performance. Patients with abdominal obesity (OR = 2.4; CI95% = 1.5–4.1), hypertension (OR = 2.2; CI95% = 1.2–3.8), diabetes (OR = 3.1; CI95% = 1.2–8.0), a longer duration of HIV infection (OR = 2.7; CI95% = 1.5–5.2), those who had AZT, DDI, D4T or DDC in their initial ART (OR = 1.9; CI95% = 1.1–3.4) and in their current ART (OR = 1.9; CI95% = 1.1–3.2) also had low SPPB performance.

In the multivariable model (Table 3), age 60 years and above (adjusted OR (aOR) = 3.4; CI95% = 1.9–5.9), being a female (aOR = 2.1; CI95% = 1.1–4.1), having an abdominal obesity (aOR = 2.1; CI95% = 1.2–4.0), a longer duration of HIV infection (aOR = 2.9; CI95% = 1.5–5.7) and AZT, DDI, D4T or DDC in their current ART (aOR = 2.0; CI95% = 1.1–3.7) remained associated with low SPPB performance.

## Discussion

In the present study, in a large sample of PLHIV aged 50 years old and above, physical function limitation evaluated with the SPPB is observed in almost 30% of the patients. The 5STS seems to be the most altered sub-test in the SPPB. PLHIV with low SPPB performance also had low unipodal balance performance, and low mean grip strength (in men only). Low SPPB performance is mainly associated with being aged above 60, but also being female, having abdominal obesity, and longer duration of the disease. The use of old Nucleoside reverse transcriptase inhibitors (NRTIs) in the current ART is also associated with lower performance.

Recent publications in older PLHIV from western countries reported similar results for the SPPB performance. In PLHIV aged 50 years or above living in the United States, it was reported a similar mean score (10.3 to 10.7) [21,22], a score similar to nearly 20-year older controls [21]. Another study in female PLHIV reported that 20% of them performed poorly with a median score of 11, but they discussed the possibility of a ceiling effect because of the participant's age in their sample (i.e. median age: 49 years (range 40–66 years)) [23]. In a cohort of drug users (median age: 51 years old), HIV infection was also associated with a 30% higher risk of reduced physical performance [24]. Functional impairment is associated with low bone and muscle mass among persons aging with HIV-infection [25]. In PLHIV but also in the general population, low SPPB performance has already been associated with poorer outcomes such as having additional comorbidities, subsequent disability, falls, higher mortality, and hospitalization rates [15,21,24,26–29], highlighting the importance of the screening of this impairment.

The most altered component of the SPPB test was the 5STS as observed in middle-aged PLHIV in western countries [9,10] and in older PLHIV [21,22]. PLHIV needed more time to complete the test than controls [22] but less than the nearly 20-year older controls [21]. As recommended in an editorial in AIDS [30], the 5STS might be the most appropriate test to evaluate physical function in the standard care of PLHIV. Our results also showed that this test was the most altered, but the absence of normative data adapted to our population was a limit to go further in our investigation. The results presented here are the baseline data of a 2-year longitudinal study. Data collected during the follow-up will enable us to investigate the evolution of the 5STS and to describe the factors which best predict this decline.

**Table 3. Analysis of factors associated with low SPPB performance in the study population.**

| Variables | Univariable model | | Multivariable model | |
|---|---|---|---|---|
| | OR (CI 95%) | p-value | aOR (CI 95%) | p-value |
| **Age** | | <0.0001* | | |
| 50–59 years old | 1 | | | |
| ≥60 years old | 2.7 (1.6–4.5) | | 3.4 (1.9–5.9) | <0.0001* |
| **Gender** | | 0.004* | | |
| Men | 1 | | | |
| Women | 2.2 (1.3–3.7) | | 2.1 (1.1–4.1) | 0.021* |
| **Marital status** | | 0.021* | | |
| In couple | 1 | | | |
| Single | 1.8 (1.1–3.0) | | | |
| **Level of education** | | 0.280 | | |
| Primary or less | 1 | | | |
| Secondary or more | 0.8 (0.5–1.2) | | | |
| **Professional activity** | | 0.001* | | |
| Employed | 1 | | | |
| Unemployed | 2.4 (1.4–3.9) | | | |
| **BMI** | | 0.339 | | |
| Normal / underweight | 1 | | | |
| Overweight/obesity | 1.3 (0.8–2.1) | | | |
| **Abdominal obesity** | | 0.001* | | |
| No | 1 | | | |
| Yes | 2.4 (1.5–4.1) | | 2.1 (1.2–4.0) | 0.014* |
| **Hypertension** (ref.: no)† | 2.2 (1.2–3.8) | 0.006* | | |
| **Hyperlipidemia** (ref.: no) | 1.9 (0.6–6.6) | 0.265 | | |
| **Diabetes** (ref.: no) | 3.1 (1.2–8.0) | 0.020* | | |
| **B or C hepatitis** (ref.: no) | 0.6 (0.1–2.9) | 0.585 | | |
| **Tuberculosis** (ref.: no) | 1.3 (0.7–2.3) | 0.420 | | |
| **Migraine** (ref.: no) | 0.7 (0.3–1.7) | 0.477 | | |
| **Arthrosis** (ref.: no) | 1.6 (0.8–3.1) | 0.165 | | |
| **Other medical problem** (ref.: no) | 1.6 (0.5–5.0) | 0.411 | | |
| **History of trauma** (ref.: no) | 0.4 (0.1–1.1) | 0.115 | | |
| **History of neurological disease** (ref.: no) | 1.6 (0.8–3.2) | 0.193 | | |
| **Duration of HIV infection (months)** | | | | |
| [7.29,88.4] | 1 | | | |
| (88.4,131] | 1.8 (0.9–3.4) | 0.071 | 1.5 (0.8–3.0) | 0.238 |
| (131,317] | 2.7 (1.5–5.2) | 0.002* | 2.9 (1.5–5.7) | 0.002* |
| **Clinical disease stages** | | | | |
| A | 1 | | | |
| B | 1.0 (0.6–1.8) | 0.997 | | |
| C | 0.8 (0.3–1.7) | 0.547 | | |
| **Nadir CD4 (cells/μl)** | | | | |
| <200 | 1 | | | |
| ≥ 200 | 1.3 (0.7–2.1) | 0.366 | | |

**Abbreviations:** aOR: Adjusted Odds Ratio, ART: Antiretroviral therapy, BMI: Body Mass Index, CI: Confidence Interval, DDC: Zalcitabine, ddI: Didanosine, D4T: Stavudine, SPPB: Short Physical Performance Battery, OR: Odd ratio, RAL: Raltégravir, AZT: Zidovudine.

*results considered as significant (p<0.05).

†ref: no: means that the OR is computed taking this category "absence of this medical problem" as a reference.

Low SPPB performance has been associated with older age. The impact of aging on physical function is well-known and linked to age-related losses in skeletal muscle mass and low aerobic capacity [31,32]. Other studies from western and African countries reported an effect of age on physical function [9–11]. We also observed that PLHIV with abdominal obesity had low physical performance whereas no association was observed with BMI. The impact of nutrition and nutritional status on physical function limitation has already been observed [9,11]. People with excessive body mass may have more limited endurance-based performance [33] and also have increased cardiovascular risks. Finally, as shown by Richert et al [9], we observed that PLHIV with longer duration of the disease were more likely to have a low physical function whereas no significant association was observed with other HIV outcomes (CD4 or viral load). The long-term effect of the virus, inflammatory process, and direct or indirect effects of ART treatment could explain this association but further investigations need to be conducted. We also observed the effect of old NRTIs included in ART, such as Zidovudine, known as having myotoxic consequences. Further studies with a more detailed protocol evaluating the duration of exposition to those molecules could help to understand better the phenomenon.

Concerning grip strength, in accordance with other studies in PLHIV or African populations [21,34–36], the mean performance was lower in women than the men ones. In a recent US study in older PLHIV, similar grip strength as in our cohort was observed in women and men (26.5 kg and 38.2 kg respectively) [21]. Weaker grip strength has also been reported in African PLHIV in comparison to HIV-uninfected subjects [37,38]. Even if men's grip strength was associated with low SPPB performance in our study, this was not the case for women. Because of the absence of normative data developed in SSA, we decided not to go any further in the interpretation of the data. As low grip strength is associated with bad outcomes (ie mortality, longer hospitalization, and physical function limitation) [39], it is important to standardize procedure [40] and to have regional specific data [41]. One study in Nigeria reported normative data for adults aged between 20 to 69 years old but with a small proportion of subjects aged ≥50 years old [34].

To our knowledge, this study represents the first opportunity to describe physical function in a large sample of older PLHIV in West Africa. However, some limitations have to be mentioned. First, we observed that the PLHIV included in the study reported few difficulties activities of daily living. The ADL and IADL are two scales usually used to evaluate the autonomy and the independence of older subjects (≥65 years old). Moreover, the questions in these scales might not be completely adapted to the West African population. Adapted scales have recently been published and might be more adapted to the African context [42]. Second, the generalizability of our results could be limited since we have included PLHIV from urban sites, under ART for at least 6 months, without important difficulties in activities of daily living and major neurologic complications or addictive behaviors. We may have selected the most adherent PLHIV and the most engaged in medical care. However, even in this population, poor physical functioning still occurs and should not be neglected. Interestingly, this population presented some specific characteristics and more specifically, more women than men were included and the majority did not report any substance use. Those characteristics could influence the generalizability of our results but also the comparison with other studies. Third, although we did not have a comparison group of HIV-negative adults included in the present study, we were able to compare our results to the prevalence of low SPPB performance reported in other papers. Fourth, the absence of normative data in grip strength adapted to our population was also a limit to go any further in some aspects of our investigation. Further studies are needed to depict this as it is an important point both for research and clinical domains. Fifth, even imputation for missing data was performed, the "viral load" variable presented some missing data, thus limiting the exploration of the association with SPPB performance.

## Conclusions

Hence, despite a successful ART, the prevalence of physical function limitation in older PLHIV is high, and this burden impacts some daily activities. Longitudinal studies should be performed to assess the evolution of the physical function limitation, their predictors, and to evaluate in more detail their consequences on daily activities. The impact of bone density, not evaluated in this study, should also be assessed. Then, how to integrate the measurement and the management of physical function limitation in the standard of care should be investigated. As the 5STS was the most altered test, it could be interesting to perform it in standard care as recommended in western countries. Efforts are now needed to develop normative data in SSA populations. Finally, age and abdominal obesity were main factors associated with low physical performance. Practitioners need to promote the practice of physical activity and to educate their patients about a better diet to limit the occurrence of overweight and obesity, and so to prevent disability in older PLHIV. Further studies are needed to develop adapted interventions focusing on a daily physical activity program, that might limit the deterioration in physical performance and the bad consequences on daily life in this population.

## Supporting information

**S1 Fig. Distribution of SPPB scores in the study population.**
(DOCX)

**S2 Fig. Distribution of scores for each SPPB subtests in the study population.**
(DOCX)

**S1 Table. Self-reported difficulties in daily physical function.**
(DOCX)

## Acknowledgments

The content is solely the responsibility of the authors and does not necessarily represent the official views of the National Institutes of Health. The IeDEA West Africa region: Site investigators and cohorts: Adult cohorts: Marcel Djimon Zannou, CNHU, Cotonou, Benin; Armel Poda, CHU Souro Sanou, Bobo Dioulasso, Burkina Faso; Fred Stephen Sarfo and Komfo Anokeye Teaching Hospital, Kumasi, Ghana; Eugene Messou, ACONDA CePReF, Abidjan, Cote d'Ivoire; Henri Chenal, CIRBA, Abidjan, Cote d'Ivoire; Kla Albert Minga, CNTS, Abidjan, Cote d'Ivoire; Emmanuel Bissagnene, and Aristophane Tanon, CHU Treichville, Cote d'Ivoire; Moussa Seydi, CHU de Fann, Dakar, Senegal; Akessiwe Akouda Patassi, CHU Sylvanus Olympio, Lomé, Togo. Pediatric cohorts: Sikiratou Adouni Koumakpai-Adeothy,_CNHU, Cotonou, Benin; Lorna Awo Renner, Korle Bu Hospital, Accra, Ghana; Sylvie Marie N'Gbeche, ACONDA CePReF, Abidjan, Ivory Coast; Clarisse Amani Bosse, ACONDA_MTCT+, Abidjan, Ivory Coast; Kouadio Kouakou, CIRBA, Abidjan, Cote d'Ivoire; Madeleine Amorissani Folquet, CHU de Cocody, Abidjan, Cote d'Ivoire; François Tanoh Eboua, CHU de Yopougon, Abidjan, Cote d'Ivoire; Fatoumata Dicko Traore, Hopital Gabriel Toure, Bamako, Mali; Elom Takassi, CHU Sylvanus Olympio, Lomé,Togo; Coordinators and data centers: François Dabis, Elise Arrive, Eric Balestre, Renaud Becquet, Charlotte Bernard, Shino Chassagne Arikawa, Alexandra Doring, Antoine Jaquet, Karen Malateste, Elodie Rabourdin, Thierry Tiendrebeogo, ADERA, Isped & INSERM U1219, Bordeaux, France. Sophie Desmonde, Julie Jesson, Valeriane Leroy, Inserm 1027, Toulouse, France. Didier Koumavi Ekouevi, Jean-Claude Azani, Patrick Coffie, Abdoulaye Cissé, Guy Gnepa, Apollinaire Horo, Christian Kouadio, Boris Tchounga, PACCI, CHU Treichville, Abidjan, Côte d'Ivoire.

## Author Contributions

**Conceptualization:** Charlotte Bernard, Aristophane Tanon, Eugène Messou, Moussa Seydi, François Dabis, Nathalie de Rekeneire.

**Formal analysis:** Charlotte Bernard, Hélène Font, Nathalie de Rekeneire.

**Funding acquisition:** François Dabis.

**Investigation:** Charlotte Bernard, Zélica Diallo, Richard Ahonon, Judicaël Malick Tine, Franklin Abouo, Nathalie de Rekeneire.

**Methodology:** Hélène Font.

**Project administration:** Charlotte Bernard, Aristophane Tanon, Eugène Messou, Moussa Seydi.

**Resources:** Zélica Diallo, Richard Ahonon, Judicaël Malick Tine, Franklin Abouo, Aristophane Tanon, Eugène Messou, Moussa Seydi.

**Software:** Hélène Font.

**Supervision:** Charlotte Bernard, Aristophane Tanon, Eugène Messou, Moussa Seydi, François Dabis, Nathalie de Rekeneire.

**Validation:** Charlotte Bernard, Hélène Font, Nathalie de Rekeneire.

**Visualization:** Charlotte Bernard.

**Writing – original draft:** Charlotte Bernard.

**Writing – review & editing:** Charlotte Bernard, Hélène Font, Zélica Diallo, Richard Ahonon, Judicaël Malick Tine, Franklin Abouo, Aristophane Tanon, Eugène Messou, Moussa Seydi, François Dabis, Nathalie de Rekeneire.

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
