## [Decision Letter · Decision Letter 0]

22 Apr 2020

PONE-D-20-01113

Prevalence and factors associated with physical function limitation in older West African people living with HIV

PLOS ONE

Dear Mrs Bernard,

Thank you for submitting your manuscript to PLOS ONE. After careful consideration, we feel that it has merit but does not fully meet PLOS ONE’s publication criteria as it currently stands. Therefore, we invite you to submit a revised version of the manuscript that addresses the points raised during the review process.

Although both reviewers felt that this was an important study, they raised some significant concerns. These included the following:

1. The data were often repeated in table and text. In some areas, this became confusing. Certain specific data items were also missing (e.g. the self-reported values for the Lickert scale) and these should be addressed.

2. The ethics committee name and the approval number should be stated

3. One reviewer expressed concerns about the absence of a negative population. Although this was acknowledged as a limitation, this does raise concerns about the conclusions. If a median population value is used for comparison, this should be adequately justified and referenced. 

In addition, both reviewers expressed that the written language required some editing

We would appreciate receiving your revised manuscript by Jun 06 2020 11:59PM. To enhance the reproducibility of your results, we recommend that if applicable you deposit your laboratory protocols in protocols.io, where a protocol can be assigned its own identifier (DOI) such that it can be cited independently in the future. For instructions see: http://journals.plos.org/plosone/s/submission-guidelines#loc-laboratory-protocols

We look forward to receiving your revised manuscript.

Kind regards,

Elizabeth S. Mayne, M.D.

Academic Editor

PLOS ONE

Journal Requirements:

3. Please include additional information regarding the survey or questionnaire to determine self-reported measures used in the study and ensure that you have provided sufficient details that others could replicate the analyses. For instance, if you developed a questionnaire as part of this study and it is not under a copyright more restrictive than CC-BY, please include a copy, in both the original language and English, as Supporting Information

'Supported by the National Institute of Mental Health (NIMH), National Cancer Institute (NCI),

the Eunice Kennedy Shriver National Institute of Child Health & Human Development

(NICHD) and the National Institute of Allergy and Infectious Diseases (NIAID) of the U.S.

National Institutes of Health (NIH), as part of the International Epidemiologic Databases to

Evaluate AIDS (IeDEA) under Award Number U01AI069919. The content is solely the

responsibility of the authors and does not necessarily represent the official views of the National

Institutes of Health. The authors have no conflicts of interest to disclose.'

'no'

Additional Editor Comments (if provided):

Although both reviewers felt that this was an important study, they raised some significant concerns. These included the following:

1. The data were often repeated in table and text. In some areas, this became confusing. Certain specific data items were also missing (e.g. the self-reported values for the Lickert scale) and these should be addressed.

2. The ethics committee name and the approval number should be stated

3. One reviewer expressed concerns about the absence of a negative population. Although this was acknowledged as a limitation, this does raise concerns about the conclusions. If a median population value is used for comparison, this should be adequately justified and referenced.

In addition, both reviewers expressed that the written language required some editing

Reviewers' comments:

Reviewer's Responses to Questions

**Comments to the Author**

1. Is the manuscript technically sound, and do the data support the conclusions?

Reviewer #1: Yes

Reviewer #2: Partly

2. Has the statistical analysis been performed appropriately and rigorously? 

Reviewer #1: Yes

Reviewer #2: I Don't Know

3. Have the authors made all data underlying the findings in their manuscript fully available?

Reviewer #1: No

Reviewer #2: No

4. Is the manuscript presented in an intelligible fashion and written in standard English?

Reviewer #1: Yes

Reviewer #2: No

5. Review Comments to the Author

Reviewer #1: The paper contributes to the body of knowledge and has relevance. There are some grammatical errors to correct and a few statements that do not read well and need to be re-written. The conclusion is adequate. The recommendations make sense, and should include further research of the co-morbidity known and mentioned - bone health / density. Tut the glaring omission is the possible interventions and actions that could be taken (and researched) to address the findings. the mainstay of this would be appropriate rehabilitation.

Reviewer #2: In this manuscript the authors evaluated 333 older HIV positive patients living in West Africa to determine the prevalence of and factors associated with physical function impairment.

Almost 30% of HIV positive patients had decreased SPPB performance which was associated with increased age, being female, having abdominal obesity and a longer disease duration. The authors suggest that physical function limitation is part of the burden of HIV disease complications of older HIV positive patients living in West Africa.

The authors must be commended on tackling such a prominent issue and generating such an abundance of data. The results are interesting however the amount of data can be overwhelming particularly as it requires better organisation, often times making it difficult to clearly see the take home message.

Major concerns

- The greatest concerns about this manuscript is the lack of a control group and/or normative data. The authors do acknowledge this however it is difficult to reach some of the conclusions that they have without comparison to an age-matched HIV-negative control group. It was also particularly worrying that grip strength was compared to an artificially constructed median point within the study population.

- As previously stated, the structure of the manuscript is sometimes difficult to follow. Too many of the tabulated results are repeated in the text. This repetition also then tends to lose some results that are only written in text. Having said that there are an abundance of results, some of the results are actually missing (or the complete set is missing). For example, where are the likert scale results for the self-reported measures? Even if this data is included as an appendix it is important that all data is presented (i.e. all questions that were asked and prevalence of answers) so that the study is reproducible.

- Quite a few statements in the abstract and the conclusion are not supported by the data/study. Particularly statements implying that HIV is impairing physical function however this cannot be concluded without an HIV negative control group. Also stating that this population is at “higher risk of falls” (line 87) and “cardiovascular risks is a factor associated with lower physical performance (line 406)” do not have supporting data.

Minor concerns

- Sometimes the language is a little unclear. The manuscript’s flow and readability would benefit from a revision.

- Line 83 (abstract): what are the values in the brackets representing and comparing?

- Overall the abstract lacks test statistics and actual test results i.e. the mean SPBB score.

- It is mentioned that trained staff performed the SPPB – how many trained staff and was inter-rated reliability confirmed?

- Please provide references for the SPPB cutoff criteria for the subtests and the CDC clinical stages.

- Overall all the table headings requires more detail i.e. sample population etc. Currently they do not stand alone.

- Table 1

- What is meant by professional activity and where are the physical activity results. It is confusing throughout

text whether “active” is referring to physical activity or professional activity.

- Units of variables e.g. years not defined.

- Suggestion- rank medical data in order of prevalence for easier reading.

- Table 2

- It is not clear that the score means are overall for the total sample and the prevalence data is for the

patients who scored below the cutoff values.

- This data would be better represented as a box and whisker plot of the data to see the distribution/spread of

the data.

- Table 3

- Table 3 is not referred to in text.

- The table heading is misleading as not all factors in the table are associated with SPPB.

- What is meant by (ref.:no)?

- Define abbreviations.

- Stats analysis

- There are some discrepancies between what is stated in the stats analysis section and how the stats analysis

appears to have been done/not done. For example in the discussion, disease duration between 50 and 60+

year olds is compared but this is not mentioned in stats analysis.

- Line 285: It is unclear what the p value is denoting as significant.

- There are missing test statistics i.e. χ2 values for χ tests.

- Were association analyses and logistic regressions only performed for patients with lower SPPB scores?

Seems inappropriate to not include the entire sample.

- Although the authors show evidence of differences between 50 and 60+ year olds, it would be helpful for them to provide a justification for splitting the ages at this specific point?

- It is inappropriate to introduce results in the discussion section. For example comparison between male and female grip strength. Additionally, comparing and discussion grip strength between men and women isn’t relevant to study.

- Overall, the discussion paragraph about grip strength doesn’t add value to the manuscript.

- Line 191 (last paragraph before stats analysis) – should be moved to study population (makes the flow of the manuscript more consistent with the results).

- Please include the ethics number and ethics committee name.

- Line 286 – who are ‘other men’ that are referred to?

6. PLOS authors have the option to publish the peer review history of their article (what does this mean?). If published, this will include your full peer review and any attached files.

Reviewer #1: Yes: Demitri Constantinou

Reviewer #2: No

---

## [Author Response · Author response to Decision Letter 0]

4 Jun 2020

Review Comments to the Author

Reviewer #1: 

The paper contributes to the body of knowledge and has relevance. 

There are some grammatical errors to correct and a few statements that do not read well and need to be re-written. The conclusion is adequate. The recommendations make sense, and should include further research of the co-morbidity known and mentioned - bone health / density. Tut the glaring omission is the possible interventions and actions that could be taken (and researched) to address the findings. the mainstay of this would be appropriate rehabilitation.

We agree with the reviewer. We corrected grammatical errors and rephrased the conclusion to add these recommendations (lines 442-459, p22).

Reviewer #2: 

In this manuscript the authors evaluated 333 older HIV positive patients living in West Africa to determine the prevalence of and factors associated with physical function impairment. 

Almost 30% of HIV positive patients had decreased SPPB performance which was associated with increased age, being female, having abdominal obesity and a longer disease duration. The authors suggest that physical function limitation is part of the burden of HIV disease complications of older HIV positive patients living in West Africa.

The authors must be commended on tackling such a prominent issue and generating such an abundance of data. The results are interesting however the amount of data can be overwhelming particularly as it requires better organisation, often times making it difficult to clearly see the take home message.

Major concerns

- The greatest concerns about this manuscript is the lack of a control group and/or normative data. The authors do acknowledge this however it is difficult to reach some of the conclusions that they have without comparison to an age-matched HIV-negative control group. It was also particularly worrying that grip strength was compared to an artificially constructed median point within the study population.

We do agree with the reviewer. This study is nested in the IeDEA West Africa cohort, a large PLHIV cohort which aims to provide resources for diverse HIV/AIDS data, without including HIV-negative adults. We considered this work as a first step as it was done in other studies without control groups (Richert et al, 2011, Baronsky et al 2014, Branas et al, 2017). We proposed to add in the limitations of the study: « even we could compare the prevalence of low SPPB performance to other published data, we were unable to include HIV negative adults. This will be important in further studies » (lines 436-439, page 21). 

Concerning grip strength, the variability in normative data between countries is important, the need to regional-specific normative data has been highlighted in a recent systematic review, and meta-analysis [38]. The variability of absolute values of grip strength is due to multiple factors, particularly health conditions and inactivity, parameters different across countries and could also be influenced by the procedure used to measure (ie table’s support, arm position, hand dominant or not) [39]. We presented grip strength here to describe the upper physical function, in complement to the description of lower physical function. The aim was not to report the prevalence of grip strength impairment but to only evaluate the association with SPPB scores. We added some descriptive information as done in the literature and also to allow comparisons with future data on this topic. According to the statistician of our team (HF), in this context, and as the median is computed from our sample data, it was the most adapted method. Finally, adding these data and discussing this point was a way to highlight the crucial need for normative data specific to West African PLHIV.

- As previously stated, the structure of the manuscript is sometimes difficult to follow. Too many of the tabulated results are repeated in the text. This repetition also then tends to lose some results that are only written in text. Having said that there are an abundance of results, some of the results are actually missing (or the complete set is missing). For example, where are the likert scale results for the self-reported measures? Even if this data is included as an appendix it is important that all data is presented (i.e. all questions that were asked and prevalence of answers) so that the study is reproducible.

We clarified this point. We modified tables 1& 2, and we added the Likert scale results in supplementary data. We checked redundancy in the text.

- Quite a few statements in the abstract and the conclusion are not supported by the data/study. Particularly statements implying that HIV is impairing physical function however this cannot be concluded without an HIV negative control group. Also stating that this population is at “higher risk of falls” (line 87) and “cardiovascular risks is a factor associated with lower physical performance (line 406)” do not have supporting data.

We rephrased the abstract and the conclusion.

Minor concerns

- Sometimes the language is a little unclear. The manuscript’s flow and readability would benefit from a revision.

The corrections have been made.

- Line 83 (abstract): what are the values in the brackets representing and comparing?

We rephrased the sentence to clarify (lines 71-72, page 3).

- Overall the abstract lacks test statistics and actual test results i.e. the mean SPBB score.

We rephrased the abstract (lines 69 – 77, p 3).

- It is mentioned that trained staff performed the SPPB – how many trained staff and was inter-rated reliability confirmed?

Before the beginning of the study, a specific training was organized, including both theoretical and practical parts. The training was conducted by NDR who is a specialist in physical function and who worked in the Health AB study and the LIFE study and certified staff study personnel for this test in the LIFE study. She was assisted by CB. We trained four doctors in Cote d’Ivoire (2 in both sites) and one doctor and one clinical research nurse in Senegal. 

During the practical part, each procedure was broken down: the trainers carrying out the procedures at the same time as the staff in training. Besides, for the same test (eg walking speed), one member of the team performed the task and the others (another person in training and the trainers) timed the time taken to do it. Then the timed times were compared. We obtained differences of less than one second on each measurement, confirming the reliability of the measures. 

- Please provide references for the SPPB cutoff criteria for the subtests and the CDC clinical stages.

Concerning SPPB cutoff criteria: Based on Guralnik’s paper (Guralnik et al, 1994), for each subtest, when the score was <4, participants reported more need for help in activities of daily living or walking ½ mile and climbing stairs increasingly across scores’ categories. In this context, we have considered that patients with a score <4 had low physical function. This reference was cited in the manuscript. 

Concerning the CDC clinical stages, we added the reference (line 145, page 6).

- Overall all the table headings requires more detail i.e. sample population etc. Currently they do not stand alone.

The corrections have been made (pages 12, 15, 17).

Table 1 :

- What is meant by professional activity and where are the physical activity results. It is confusing throughout text whether “active” is referring to physical activity or professional activity.

We corrected this confusing point. Professional activity was defined in two categories « employed/unemployed ». Physical activity results are now presented in the Table 1 (pages 12-13) and in a supplementary table (S1 Table) for the Likert scale.

- Units of variables e.g. years not defined.

- Suggestion- rank medical data in order of prevalence for easier reading.

The corrections have been made (pages 12-13).

Table 2

- It is not clear that the score means are overall for the total sample and the prevalence data is for the patients who scored below the cutoff values.

- This data would be better represented as a box and whisker plot of the data to see the distribution/spread of the data.

We decided to add in Table 1 (pages 12-13) the percentage of the maximum score for ADL and IADL and also to add mean score for each test to complete the description of the sample. In supplementary data, we also added histograms to describe in detail the SPPB scores (S1 and S2 Fig.) and the description of the Likert scale in supplementary data (S1 Table). To clarify, Table 2 now only concerns the prevalence of low performance for each score (page 15).

Table 3

- Table 3 is not referred to in text.

- The table heading is misleading as not all factors in the table are associated with SPPB.

- What is meant by (ref.:no)?

- Define abbreviations.

The corrections have been made (lines 336 & 343, page 16 + page 17).

Stats analysis

- There are some discrepancies between what is stated in the stats analysis section and how the stats analysis appears to have been done/not done. For example in the discussion, disease duration between 50 and 60+ year olds is compared but this is not mentioned in stats analysis.

We clarified this paragraph with adding information (lines 221-225, page 9). 

Concerning the sentence in the discussion: the comparison of disease duration between 50 and 60+ year old was just to discuss the fact that the association between low SPPB performance and duration of the disease was not due to the age of the PLHIV. But as it was confusing, we removed the sentence (lines 398-400, page 20). 

- Line 285: It is unclear what the p value is denoting as significant.

This point was clarified by adding a sentence: « A significant association between SPPB score and unipodal test performance was observed (p=0.001).” (lines 322-323, page 15)

- There are missing test statistics i.e. χ2 values for χ tests.

We added these informations in the text (lines 329-333, page 16).

- Were association analyses and logistic regressions only performed for patients with lower SPPB scores? Seems inappropriate to not include the entire sample.

Logistic regression analyses were performed in the whole sample. We described factors associated with low SPPB performance.

- Although the authors show evidence of differences between 50 and 60+ year olds, it would be helpful for them to provide a justification for splitting the ages at this specific point?

In another study conducted by our team (Bernard et al, 2018, DOI : 10.2147/HIV.S172198), we observed that PLHIV aged above 60 years old has poorer outcomes than the ones aged between 50 to 59 years old. They had an increased risk of mortality and lost to follow-up. In this context, we splitted the ages at this specific point.

- It is inappropriate to introduce results in the discussion section. For example comparison between male and female grip strength. Additionally, comparing and discussion grip strength between men and women isn’t relevant to study. 

- Overall, the discussion paragraph about grip strength doesn’t add value to the manuscript.

We added the mean performance in the discussion because we discussed them with the data available in the literature. It was to facilitate the reading (not necessary to back to the results section to compare the values with the one in the literature). But we agree that it could be considered inappropriate so we removed this (lines 408-410, page 20). 

Concerning the discussion on grip strength: as normative data are established according to gender, comparing men and women is often done and this comparison is often discussed in the literature. Unfortunately, we could not go further in the description of the data because of the absence of normative data in the West African population. As scarce data on grip strength are available in SSA, discussing this point was a mean to highlight this point. In this context, we decided to keep this paragraph in the discussion.

- Line 191 (last paragraph before stats analysis) – should be moved to study population (makes the flow of the manuscript more consistent with the results).

We moved this paragraph in the study population section (lines 133-149, page 6).

- Please include the ethics number and ethics committee name.

We added this information (lines 127-129, page 5).

- Line 286 – who are ‘other men’ that are referred to?

We clarified this point in the text (line 325, page 15).

---

## [Decision Letter · Decision Letter 1]

27 Jul 2020

PONE-D-20-01113R1

Prevalence and factors associated with physical function limitation in older West African people living with HIV

PLOS ONE

Dear Dr. Bernard,

Thank you for submitting your manuscript to PLOS ONE. After careful consideration, we feel that it has merit but does not fully meet PLOS ONE’s publication criteria as it currently stands. Therefore, we invite you to submit a revised version of the manuscript that addresses the points raised during the review process.

This revision is significantly improved but a number of minor modifications have been suggested. These are included under reviewer 2's suggestions. We would also recommend English language editing,

We look forward to receiving your revised manuscript.

Kind regards,

Elizabeth S. Mayne, M.D.

Academic Editor

PLOS ONE

Journal Requirements:

Additional Editor Comments (if provided):

The reviewers felt that this article was significantly improved but a number of minor alterations have been suggested including:

1. Some English Language editing

2. Clarification of some terminology utilised including, for example, concepts of adherence, the concept of a "recent" CD4+ T cell count. Where data are missing, this should be indicated as a limitation

3. In some cases, there should be clarification of the definitions of physical limitation and also of severe disability. Where there are overlapping categories, it should be indicated (this includes substance abuse)

4. Table 3 requires modification especially to include variables like marital status

Reviewers' comments:

Reviewer's Responses to Questions

**Comments to the Author**

1. If the authors have adequately addressed your comments raised in a previous round of review and you feel that this manuscript is now acceptable for publication, you may indicate that here to bypass the “Comments to the Author” section, enter your conflict of interest statement in the “Confidential to Editor” section, and submit your "Accept" recommendation.

Reviewer #1: All comments have been addressed

Reviewer #3: (No Response)

2. Is the manuscript technically sound, and do the data support the conclusions?

Reviewer #1: Yes

Reviewer #3: Yes

3. Has the statistical analysis been performed appropriately and rigorously? 

Reviewer #1: Yes

Reviewer #3: Yes

4. Have the authors made all data underlying the findings in their manuscript fully available?

Reviewer #1: Yes

Reviewer #3: Yes

5. Is the manuscript presented in an intelligible fashion and written in standard English?

Reviewer #1: Yes

Reviewer #3: Yes

6. Review Comments to the Author

Reviewer #1: (No Response)

Reviewer #3: This is an interesting article that deals with an understudied area of HIV management. The researchers produced important data which have implications for the clinical management of patients and are worthy of publication. The manuscript has already gone through one round of review. I found the comments made by reviewer 2 particularly pertinent and thorough but am of the opinion that the researchers have adequately addressed the concerns that had been raised. As is inevitable with a second round of review by a new reviewer, I do, however, have additional comments and questions for the researchers.

Lines 145 – 147: “Exposure to raltégravir (RAL), zidovudine (AZT), didanosine (ddI), stavudine (D4T), zalcitibine (DDC) in the initial ART treatment and also in the current ART

treatment was studied through a categorical variable (yes/no).” What is the reason for including RAL with the old NRTIs which are known to have significant toxicity and may be confounded by a longer duration of treatment? RAL is a different class (INSTI), has limited toxicity and has not been available for as long as the NRTIs listed, so would be expected to perform differently. [Please note that zalcitabine should be zalcitabine].

Lines 147 -149: The researchers should justify the selection of and provide a reference for their definition of adherence: “Adherence to ART was defined as the percentage of tablets the patient declared to have taken over 7 days (in comparison to the prescribed total number of tablets over this period).”

Line 170: “A ‘physical function impairment’ variable was also described including at least more than one test altered, as defined above.” How was this variable determined or what was it based on?

The statistics section is thorough and commendable. My only comment on this section is that there is no mention about whether the researchers had assessed for confounding or effect modification. This is especially important in case of evaluating the effect of the various antiretroviral agents – see later comment.

Line 251: “Almost half of them lived as a couple (46.3%)”. I would rather focus on the majority who lived alone, especially since this has been reported as an independent factor related to frailty, at least in men – see Kojima et al. Is living alone a risk factor of frailty? A systematic review and meta-analysis. Ageing Research Reviews Volume 59, May 2020, 101048. https://doi.org/10.1016/j.arr.2020.101048

Table 1:

The denominator in the section Anthropometric and medical data should be given since, judging by the percentages, they do all seem to be 333.

Clinical disease stages at ART initiation – specify that these are CDC stages.

Where data are missing (e.g. for nadir and more recent CD4 count) this should be indicated as such. What was regarded as a “more recent CD4” count?

Detectable Viral load – what was the viral load cut-off used for this variable?

“Poor Observance” should rather be poor adherence

To make sense of the comparison between antiretroviral medication, the other drugs should also be listed.

Something went wrong with the alignment of the second part of Table 1 – please correct.

Line 270: Arthrosis is a very non-specific term. Could the authors please explain what they mean by it?

Line 277: Few patients reported substance use (<8%). Since this cannot be understood from the table, where alcohol and drug use are reported separately and make up 9.02%, and since some people therefore seemingly overlap, the numbers adding up to this conclusion should be provided.

Table 2 is now much improved and easier to understand.

Lime 287 – 289: “Few patients reported moderate to severe difficulties in daily physical function (<4%), except for climbing stairs (14.1%) and carrying a grocery bag (8.4%). Among patients reporting those difficulties, less than 2.5% reported severe or extreme difficulties.” Overlapping categories (namely severe) make this difficult to follow.

Table 3 needs to be reworked to ensure alignment between the p-values in the various rows. It is not clear why all the variables meeting the inclusion criteria for entry into the multivariable model (e.g. marital situation; professional activity; hypertension; diabetes; CD4 count, etc) are not presented in this model.

Why are some p-values in the table underlined?

Some abbreviations are incorrect: ART is antiretroviral therapy and not just antiretroviral; CI is confidence interval and not confident interval.

Lines 362 – 363: “Physical function limitation seems to be more important for PLHIV aged above 60 years old, in women, for those with abdominal obesity and longer duration of the disease.” An association is not the same as importance.

Lines 363 - 364: “The use of certain molecules at the initial ART is also associated to lower performance.” This sentence has been deleted but the variable was found to be significantly associated with SPPB performance in the results. It is not clear why is has been deleted and why it is also not listed as a significant association in the abstract or discussion. However, before including this variable in the discussion, it should be clear whether it had been assessed for confounding e.g. by duration of treatment? In addition, could the researchers determine which component of SPPB performance was most affected by ART? This could inform their future work.

Line 371: “but they discussed the possibility of a ceiling effect in their sample.” The ceiling effect should be better explained.

Line 422: Abbreviations that have been introduced, such as ART, should be used throughout.

The language is greatly improved, but there are still a few sentences that should be corrected. To name but a few:

Abstract: line 62: “older ones” should rather be something like “older people”.

Abstract: line 70: “putting this population at disability” – should this be at risk for disability?

Line 113: “type-1 HIV” should be written as HIV-1.

Line 160: “following the standard procedures provided in Guralnik et al. paper”

Line 168: “If the patient could not stand on one leg 30 seconds…”

Matrimonial (Table 1) or marital situation (Table 3) should rather been marital status.

Line 436 - 437: “even we could compare the prevalence of low SPPB performance to other published data…”

7. PLOS authors have the option to publish the peer review history of their article (what does this mean?). If published, this will include your full peer review and any attached files.

Reviewer #1: No

Reviewer #3: No

---

## [Author Response · Author response to Decision Letter 1]

9 Sep 2020

Dear Editor-in-Chief, 

Please find below the answers to the comments of the referees, point by point. In the revised manuscript, we highlighted the changes by using the track changes. Here, we indicate the corresponding lines and pages in the track manuscript of the changes.

Editor Comments:

The reviewers felt that this article was significantly improved but a number of minor alterations have been suggested including:

1. Some English Language editing

We corrected those points.

2. Clarification of some terminology utilised including, for example, concepts of adherence, the concept of a "recent" CD4+ T cell count. Where data are missing, this should be indicated as a limitation

We clarified the terminology. We added the presence of missing data as a limitation (lines 409-411, p. 21). 

3. In some cases, there should be clarification of the definitions of physical limitation and also of severe disability. Where there are overlapping categories, it should be indicated (this includes substance abuse).

We clarified this point. 

4. Table 3 requires modification especially to include variables like marital status.

This Table was updated.

Review Comments to the Author

Reviewer #3: 

This is an interesting article that deals with an understudied area of HIV management. The researchers produced important data which have implications for the clinical management of patients and are worthy of publication. The manuscript has already gone through one round of review. I found the comments made by reviewer 2 particularly pertinent and thorough but am of the opinion that the researchers have adequately addressed the concerns that had been raised. As is inevitable with a second round of review by a new reviewer, I do, however, have additional comments and questions for the researchers.

Lines 145 – 147: “Exposure to raltégravir (RAL), zidovudine (AZT), didanosine (ddI), stavudine (D4T), zalcitibine (DDC) in the initial ART treatment and also in the current ART treatment was studied through a categorical variable (yes/no).” What is the reason for including RAL with the old NRTIs which are known to have significant toxicity and may be confounded by a longer duration of treatment? RAL is a different class (INSTI), has limited toxicity and has not been available for as long as the NRTIs listed, so would be expected to perform differently. [Please note that zalcitabine should be zalcitabine].

Among the side effects of RAL, functional disorders have been cited (dizziness in posture, peripheral neuropathy, paresthesia). For this reason, we initially decided to include this molecule in this categorical variable. In our sample, RAL was included in the current ART combination for 4 patients. No patient got RAL in their initial ART combination. So, few patients got RAL in their ART and none reported side effects. In this context and based on the reviewer’s comment, we decided to re-run the analysis without RAL in the variable. The results remain significant. The results (lines 307-328, p. 16), Table 3 (p. 17), and the Abstract are updated. We corrected the spelling in DDC in the text.

Lines 147 -149: The researchers should justify the selection of and provide a reference for their definition of adherence: “Adherence to ART was defined as the percentage of tablets the patient declared to have taken over 7 days (in comparison to the prescribed total number of tablets over this period).”

We used the clinical definition of ART adherence used in the local pharmacies located at the clinical sites where patients were included.

Line 170: “A ‘physical function impairment’ variable was also described including at least more than one test altered, as defined above.” How was this variable determined or what was it based on?

We clarified the sentence in the manuscript (lines 164-167, p. 8). This variable is a multidomain variable. A patient was considered as having physical function impairment if at least more than one test (i.e. balance, gait speed, 5-STS or unipodal balance test) was altered. For each test, we detailed in the Methods the cut-off used to define when the test was considered as altered (p. 8). 

The statistics section is thorough and commendable. My only comment on this section is that there is no mention about whether the researchers had assessed for confounding or effect modification. This is especially important in case of evaluating the effect of the various antiretroviral agents – see later comment.

We used a backward selection, an automatic process that selects the model the best fitted to explain the dependent variable (i.e. altered or not SPPB performance). In this automatic process, collinearities and effect modifications are assessed automatically. “Inclusion centers” variable was included as a cofounder in each model

We checked for the interaction between ART treatment and duration. As the interaction was found as non-significant, we did not include it in the final model. 

Line 251: “Almost half of them lived as a couple (46.3%)”. I would rather focus on the majority who lived alone, especially since this has been reported as an independent factor related to frailty, at least in men – see Kojima et al. Is living alone a risk factor of frailty? A systematic review and meta-analysis. Ageing Research Reviews Volume 59, May 2020, 101048. https://doi.org/10.1016/j.arr.2020.101048

We rephrased the sentence (lines 229-230, p. 11).

Table 1: The denominator in the section Anthropometric and medical data should be given since, judging by the percentages, they do all seem to be 333.

Clinical disease stages at ART initiation – specify that these are CDC stages.

Where data are missing (e.g. for nadir and more recent CD4 count) this should be indicated as such. What was regarded as a “more recent CD4” count?

Detectable Viral load – what was the viral load cut-off used for this variable?

“Poor Observance” should rather be poor adherence

To make sense of the comparison between antiretroviral medication, the other drugs should also be listed. Something went wrong with the alignment of the second part of Table 1 – please correct.

From the comments, we have updated and reformatted the Table 1 (p. 12-13).

We corrected the percentages in the Table 1. In the old version, the denominators varied depending on the number of missing data. Now, all the percentages are calculated using one denominator (N=333), and missing data are mentioned. We also added the presence of missing data as a limitation (lines 409-411, p. 21). 

The most recent CD4 is the last measure of the CD4 available in the patient’s medical record.

The viral load cut-off was 50 copy/ml of blood (N=143). For 73 patients, the laboratory detection threshold was different (viral load was considered as undetectable when viral load was <100 copy/ml (N=69), or <200 copy/ml (N=1) or <300 copy/ml (N=3)). As the laboratory did, we considered these patients as having an undetectable viral load at these thresholds. 

Line 270: Arthrosis is a very non-specific term. Could the authors please explain what they mean by it?

We used a medical definition of arthrosis. Arthrosis was defined as chronic and persistent pain in the joints. This could be caused by abnormal wear and tear of the cartilage and the entire joint. As we tested physical function and grip strength, this information was essential to identify patients with this condition, with a higher risk of pain to realize the tests. 

Line 277: Few patients reported substance use (<8%). Since this cannot be understood from the table, where alcohol and drug use are reported separately and make up 9.02%, and since some people therefore seemingly overlap, the numbers adding up to this conclusion should be provided.

By <8%, we wanted to report that less than 8% were hazardous drinkers or reported drug use. Only two patients were both hazardous drinkers and drug users.

Table 2 is now much improved and easier to understand.

Lime 287 – 289: “Few patients reported moderate to severe difficulties in daily physical function (<4%), except for climbing stairs (14.1%) and carrying a grocery bag (8.4%). Among patients reporting those difficulties, less than 2.5% reported severe or extreme difficulties.” Overlapping categories (namely severe) make this difficult to follow.

We clarified this point (lines 270-273, p. 14).

Table 3 needs to be reworked to ensure alignment between the p-values in the various rows. It is not clear why all the variables meeting the inclusion criteria for entry into the multivariable model (e.g. marital situation; professional activity; hypertension; diabetes; CD4 count, etc) are not presented in this model. Why are some p-values in the table underlined? Some abbreviations are incorrect: ART is antiretroviral therapy and not just antiretroviral; CI is confidence interval and not confident interval.

We reworked Table 3 (p. 17-18).

Unbalanced variables (85%/15%) were excluded from the multivariable analysis. The final model was obtained with an automatic backward selection, and we considered significant associations at p < 0.05. This procedure is mentioned in the methods (lines 217-220, p 10). The statistician working in our team (HF) validated the procedure.

We corrected the abbreviations.

Lines 362 – 363: “Physical function limitation seems to be more important for PLHIV aged above 60 years old, in women, for those with abdominal obesity and longer duration of the disease.” An association is not the same as importance.

We rephrased this sentence (lines 332-335, p 18).

Lines 363 - 364: “The use of certain molecules at the initial ART is also associated to lower performance.” This sentence has been deleted but the variable was found to be significantly associated with SPPB performance in the results. It is not clear why is has been deleted and why it is also not listed as a significant association in the abstract or discussion. However, before including this variable in the discussion, it should be clear whether it had been assessed for confounding e.g. by duration of treatment? In addition, could the researchers determine which component of SPPB performance was most affected by ART? This could inform their future work.

We re-added this sentence in the Discussion section and added a sentence in the abstract. The association between this variable and SPPB performance is discussed in the Discussion section (lines 370-373, p 20). 

As explained above, we used a backward selection. In this automatic process, collinearities and effect modifications are assessed automatically. We also checked for the interaction between ART treatment and disease duration. As the interaction was found as non-significant, we did not include it in the final model.

As suggested by the Reviewer and for information, we evaluated which component of SPPB was most affected by ART. It appeared that the ART variable was only associated with STS. However, even if it could be very interesting, we decided not to add this information in the manuscript. Indeed, we think that it would require more details, such as specific multivariate models to control that ART is associated with each part considering all other potential covariates. Furthermore, reviewers / readers could ask why we do not detail which component of the SPPB is affected by each factor associated with low SPPB performance. This level of detail does not seem to be under the scope of our paper.

Line 371: “but they discussed the possibility of a ceiling effect in their sample.” The ceiling effect should be better explained.

As SPPB is designed for older subjects, the ceiling effect was linked to participants' age, younger than our patients. We rephrased the sentence and added this information: “they discussed the possibility of a ceiling effect because of the participant’s age in their sample (i.e., median age: 49 years (range 40–66 years))” (lines 340-341, p.19).

Line 422: Abbreviations that have been introduced, such as ART, should be used throughout.

We corrected this point.

The language is greatly improved, but there are still a few sentences that should be corrected. To name but a few:

Abstract: line 62: “older ones” should rather be something like “older people”.

Abstract: line 70: “putting this population at disability” – should this be at risk for disability?

Line 113: “type-1 HIV” should be written as HIV-1.

We corrected those points.

Line 160: “following the standard procedures provided in Guralnik et al. paper”

We rephrased this part of the sentence (line 154, p 8).

Line 168: “If the patient could not stand on one leg 30 seconds…”

We rephrased this part of the sentence (lines 162-163, p 8).

Matrimonial (Table 1) or marital situation (Table 3) should rather been marital status.

We corrected this point using “marital status” in Tables 1 & 3.

Line 436 - 437: “even we could compare the prevalence of low SPPB performance to other published data…”

 We rephrased the sentence (lines 404-406, p 21).

---

## [Decision Letter · Decision Letter 2]

6 Oct 2020

Prevalence and factors associated with physical function limitation in older West African people living with HIV

PONE-D-20-01113R2

Dear Dr. Bernard,

We’re pleased to inform you that your manuscript has been judged scientifically suitable for publication and will be formally accepted for publication once it meets all outstanding technical requirements.

Kind regards,

Elizabeth S. Mayne, M.D.

Academic Editor

PLOS ONE

Additional Editor Comments (optional):

Reviewers' comments:

Reviewer's Responses to Questions

**Comments to the Author**

1. If the authors have adequately addressed your comments raised in a previous round of review and you feel that this manuscript is now acceptable for publication, you may indicate that here to bypass the “Comments to the Author” section, enter your conflict of interest statement in the “Confidential to Editor” section, and submit your "Accept" recommendation.

Reviewer #3: All comments have been addressed

2. Is the manuscript technically sound, and do the data support the conclusions?

Reviewer #3: Yes

3. Has the statistical analysis been performed appropriately and rigorously? 

Reviewer #3: Yes

4. Have the authors made all data underlying the findings in their manuscript fully available?

Reviewer #3: Yes

5. Is the manuscript presented in an intelligible fashion and written in standard English?

Reviewer #3: Yes

6. Review Comments to the Author

Reviewer #3: There are a few minor, and mostly editorial issues, that should be addressed but do not require re-review:

Line 256: “Concerning ART, 68.2% had AZT, DDI, DAT, DDC included in their initial treatment” – DAT should be d4T.

Lines 258-9: “Few patients reported substance use (hazardous drinkers or drug users <8%, except 17.7% for tobacco users (current/previous)).” Inside brackets should be square.

Small issues:

DDI should be ddI

D4T should be d4T

DDC should be ddC

Raltégravir should be raltegravir

Nucleoside reverse transcriptase inhibitors should not be capitalised.

Line 410-2: This sentence should be rewritten for clarity: “Fifth, even imputation for missing data was performed, the “viral load” variable presented some missing data, thus limiting the exploration of the association with SPPB performance”.

Table 1:

there are some numbers in the background that are overlapping with the text.

More recent CD4 should be Most recent CD4

I am satisfied with the explanation of how “detectable viral load” had been defined and think it would be helpful to add this as a footnote at the end of Table 1.

A few language errors remain and the manuscript will be improved by careful language editing. For instance, "on ART since ≥6 months" should be on ART for ≥6 months; "having an abdominal obesity" should just be abdominal obesity; "With aging, alteration of physical function ... are more common" should be is more common; etc.

7. PLOS authors have the option to publish the peer review history of their article (what does this mean?). If published, this will include your full peer review and any attached files.

Reviewer #3: No

---

## [Editor Report · Acceptance letter]

13 Oct 2020

PONE-D-20-01113R2 

Prevalence and factors associated with physical function limitation in older West African people living with HIV 

Dear Dr. Bernard:

I'm pleased to inform you that your manuscript has been deemed suitable for publication in PLOS ONE. Congratulations! Your manuscript is now with our production department. 

Kind regards, 

on behalf of

Dr. Elizabeth S. Mayne 

Academic Editor

PLOS ONE